# Multiple reaction monitoring assays for large-scale quantitation of proteins from 20 mouse organs and tissues

Sarah A. Michaud [1✉], Helena Pětrošová[1], Nicholas J. Sinclair[1], Andrea L. Kinnear[1], Angela M. Jackson[1], Jamie C. McGuire[1], Darryl B. Hardie [1], Pallab Bhowmick [1], Milan Ganguly[2,3], Ann M. Flenniken [2,4], Lauryl M. J. Nutter [2,3], Colin McKerlie[3], Derek Smith[1], Yassene Mohammed[5,6,7], David Schibli[1], Albert Sickmann[6] & Christoph H. Borchers [7,8,9,10✉]

Mouse is the mammalian model of choice to study human health and disease due to its size, ease of breeding and the natural occurrence of conditions mimicking human pathology. Here we design and validate multiple reaction monitoring mass spectrometry (MRM-MS) assays for quantitation of 2118 unique proteins in 20 murine tissues and organs. We provide open access to technical aspects of these assays to enable their implementation in other laboratories, and demonstrate their suitability for proteomic profiling in mice by measuring normal protein abundances in tissues from three mouse strains: C57BL/6NCrl, NOD/SCID, and BALB/cAnNCrl. Sex- and strain-specific differences in protein abundances are identified and described, and the measured values are freely accessible via our MouseQuaPro database: http://mousequapro.proteincentre.com. Together, this large library of quantitative MRM-MS assays established in mice and the measured baseline protein abundances represent an important resource for research involving mouse models.

[1] University of Victoria-Genome British Columbia Proteomics Centre, Victoria, BC, Canada. [2] The Center for Phenogenomics, Toronto, ON, Canada. [3] The Hospital for Sick Children, Toronto, ON, Canada. [4] Sinai Health Lunenfeld-Tanenbaum Research Institute, Toronto, ON, Canada. [5] Center for Proteomics and Metabolomics, Leiden University Medical Center, Leiden, the Netherlands. [6] Leibniz-Institut für Analytische Wissenschaften-ISAS-e.V, Dortmund 44139, Germany. [7] Segal Cancer Proteomics Centre, Lady Davis Institute, Jewish General Hospital, McGill University, Montreal, QC, Canada. [8] Gerald Bronfman Department of Oncology, Jewish General Hospital, Montreal, QC, Canada. [9] Department of Experimental Medicine, McGill University, Montreal, QC, Canada. [10] Department of Pathology, McGill University, Montreal, QC, Canada. ✉email: sarah@proteincentre.com; christoph.borchers@mcgill.ca

**M**us musculus has long served as a model organism for studying both normal biological processes and mechanisms underlying disease. Although the challenges of direct translation from mouse models to human biology and disease are well known[1–3], several advantages make mice indispensable to the study of human health including genetic similarity and availability of tools for molecular manipulation[4,5]. Numerous genetically engineered mouse strains (e.g., knockout, knock-in, and transgenic) have been designed and generated to study gene function and model human diseases[6]. To fully characterize the phenotype of such strains, multiple tissues and organ systems must be examined[7–10] at the molecular level including protein expression[11,12]. Mass spectrometry (MS) is the primary method used to quantify tissue proteomes due its ability to achieve precise, reproducible, and robust measurements of proteins with high throughput[13–15]. Several different MS techniques exist, and can broadly be grouped into untargeted "discovery" methods, which aim to identify as many proteins as possible from a sample, or targeted methods that analyze a pre-defined set of analytes with high specificity and sensitivity[15–19]. Recent advancements in untargeted proteomic methods allow their use for large-scale protein quantitation, however, targeted methods including multiple reaction monitoring MS (MRM-MS) remain the gold standard in the field. This is due to the high sensitivity, selectivity, and robustness of MRM-MS that is achieved via use of stable isotope-labeled standards[20–26]. Implementing standards allows to control for sample-specific ionization effects, including ion suppression and presence of interferences that affect quantitation, and ensures reproducibility across different laboratories and points of time[27–29]. MRM-MS assays should be therefore implemented when precise and reproducible quantitation of protein abundance is required.

Quantitative MRM-MS assays are typically developed using one or more proteotypic peptides as surrogates for a protein target and are optimized and validated through a multi-step process. Best practice for the development of such assays has been discussed at length in the proteomic community, covering topics such as selection, use, and handling of labeled standards[30–32], calibration strategy[33–36], and experimental steps of the development and validation process[28,37,38]. In general, the steps for assay development and validation are both time consuming and costly. As a result, only a few reports of large-scale assay development exist[39,40], with most MRM analyses focusing on less than 100 analytes.

In this work we aimed to reduce the barriers to implementing high-quality MRM-MS assays by the broader scientific community via (a) providing details on optimized and validated MRM-MS assays that are ready-to-use upon obtaining the respective peptide standards and (b) providing expected values of the corresponding protein abundances across commonly used mouse models. To accomplish this, we developed 7184 quantitative MRM-MS assays that measure 2118 unique proteins across 20 mouse organs and tissues. Our group previously described the development of 500 MRM-MS assays in mouse plasma[40], and here we extend this work by developing assays for brain, eye, salivary gland, heart, lung, liver caudate and right lobe, liver left

lobe, pancreas, spleen, kidney, ovary, testis, epididymis, seminal vesicle, skin, skeletal muscle tissue, brown and white adipose tissue, and blood separated into plasma and red blood cell portions. Development and validation was performed using heavy-labeled peptide standards according to the Clinical Proteome Tumor Analysis Consortium (CPTAC) guidelines[37,41] and we provide the parameters of these assays to the community as a resource. We demonstrate the significance and applicability of this large cohort of validated assays for the molecular phenotyping of mice by measuring protein concentrations in samples from six male and six female mice of three common mouse laboratory strains: C57BL/6NCrl, NOD/SCID, and BALB/cAnCrl. Across all organs and tissue types a total of 5149 concentration measurements were obtained, corresponding to 1691 proteins. Both strain- and sex-specific differences in protein expression were observed for the various organs and tissues. Overall, these measurements advance our knowledge of normal protein concentrations in mice, provide important considerations for experimental design involving mouse models, and demonstrate the power of MRM-MS assays for the study of complex biological processes.

## Results and discussion
We describe the development and application of quantitative MRM-MS assays for 20 mouse organs and tissues achieved through: (1) identification of proteins and peptides in each organ or tissue type, (2) selection of protein and peptide targets for assay development, (3) development of quantitative MRM-MS assays using stable isotope-labeled standards, and (4) grouping the assays into panels and subsequent measurement of sample protein concentrations. An overview of the numbers of peptides and corresponding unique proteins at each stage are provided in Table 1, and are further described in the following sections. A detailed summary is found in Supplementary Data 1.

**Identification of proteins and peptides in each organ or tissue type**. To determine proteins detectable in each organ and tissue type, pooled samples from female and male C57BL/6NCrl mice were analyzed by untargeted MS on an Orbitrap Fusion Tribrid instrument. Previously, Geiger et al. identified 7349 proteins by the analysis of 28 tissues from SILAC-labeled mice[42]. Our analysis identified a comparable total of 5033 unique proteins across 20 organ and tissue types from 54,701 peptide sequences (Fig. 1; specific peptides and proteins identified in each sample type are listed in Supplementary Data 2 and 3). The number of proteins identified ranged 10-fold from approximately 200 proteins identified from blood (187 proteins in plasma and 216 proteins in red blood cells) to >2000 proteins identified from brain and testis (2126 and 2316 proteins, respectively). The discovery proteomics workflow did not include fractionation, enrichment, or depletion, which are commonly used to increase the depth of proteome coverage at the cost of time and reproducibility[43–49]. We avoided these strategies in order to match the sample preparation used for MRM-MS.

**Table 1 Number of peptides and proteins evaluated at each stage of assay development and sample measurement.**

|  | (1) Identification of proteins and peptides | (2) Selection of protein and peptide targets | (3) Development of quantitative assays | (4) Measurement of sample protein concentrations |
|---|---|---|---|---|
| No. Peptides | 54,701 | 2965 | *7184 | *5149 |
| No. Proteins | 5033 | 2675 | 2118 | 1691 |

Numbers represent totals of unique peptides or proteins, except those indicated by an asterisk (*) which represent the sum of unique peptides in each tissue.

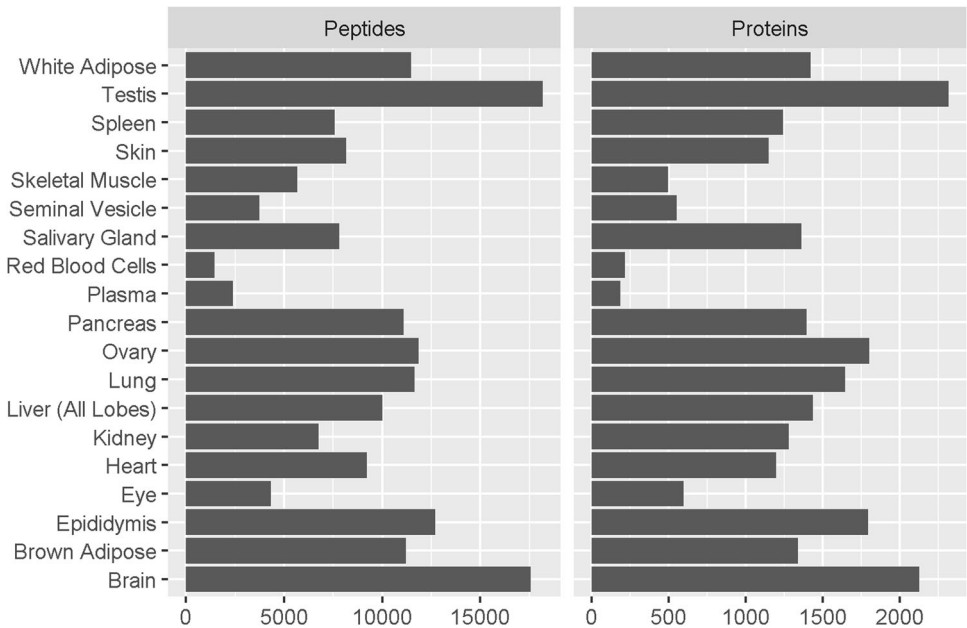

**Fig. 1 Number of peptides and proteins identified by untargeted MS analysis of 20 mouse organs and tissues.** Pooled samples from $n = 3$ females and $n = 3$ males were analyzed on an Orbitrap Fusion instrument. In total 5033 unique proteins were identified across all 20 organs and tissue types. Search results were filtered to a 1% false discovery rate and a minimum of two identified peptides per protein.

**Selection of protein and peptide targets for assay development**.
With the exception of red blood cells and plasma, the number of proteins identified by the untargeted analysis was too high to allow assay development for all detected targets. Proteins were therefore selected using criteria described in Material and Methods and limited to 700 or less per organ or tissue. In some cases, assays were developed for proteins with known involvement in disease that were not detectable in normal, healthy mice. For example, the protein Bridging integrator 2 (D3Z6Q9), which is associated with neuroinflammation and Alzheimer's disease[50,51], was not detected in healthy brain tissue samples but might be upregulated in a disease model. Similarly, one third of plasma assays targets proteins are not detectable in healthy mice. Plasma is an important biofluid for disease detection and monitoring[52,53], and these proteins can also serve as indicators of disease. All biologically-relevant information on the selected proteins can be found in our MouseQuaPro database (http://mousequapro.proteincentre.com)[54] developed for this purpose.

In total, 2675 proteins were selected for assay development. For each target protein, one to three proteotypic peptide surrogates were selected using PeptidePicker software[32] and the corresponding stable isotope-labeled heavy and unlabeled light peptide standards were synthesized in-house. Standards were successfully synthesized for 2965 unique peptide sequences (Supplementary Data 4), which ranged in length from 6 to 25 amino acids (Fig. 2a) and had an average HPLC (high-performance liquid chromatography) retention time of 19 min (Fig. 2b). Peptides were slightly hydrophilic with an average GRAVY (grand average of hydropathy index) score of −0.21 (Fig. 2c). Optimal acquisition parameters, including dominant precursor charge state (Fig. 2d), most intense fragment ions (Fig. 2e), and optimal instrument collision energy were determined experimentally for each sequence to increase the sensitivity of detection.

**Development of quantitative MRM-MS assays**. Each MRM-MS assay underwent rigorous characterization and validation according to the guidelines set out by the Clinical Proteomic Tumor Analysis Consortium (CPTAC) working group[41], which

unifies assay development across the proteomics community and ensures the quality of each assay. The validation included generation of the peptide response curve to determine the assay's lower limit of quantitation (LLOQ) and linear range and determination of the assay's repeatability[40]. In all validation experiments, heavy-labeled peptide standards were spiked into a representative sample matrix for each organ or tissue type, pooled from three male and three female C57BL/6NCrl mice (see Material and Methods for details).

To generate the response curve, pooled sample matrix was spiked with heavy-labeled peptide standards ranging in concentration from 20,000 to 0.3125 fmol. Spiked samples were injected in triplicate, and heavy to light signal ratios were determined. The linear range was defined as the concentrations for which the mean peak area ratio was within ±20% of the expected concentration. The LLOQ of the assay was defined as the lowest concentration within the linear range where the coefficient of variation (CV) was less than 20%[40]. Assays which did not meet these criteria were excluded from further development (Fig. 3a and Supplementary Data 1). In parallel, the concentrations of the endogenous peptides were approximated by single point measurement using the heavy peptide spiked into the pooled sample matrix. If the concentration of the endogenous peptide was greater than 500x LLOQ, the LLOQ was adjusted upwards within the assay's linear range to ensure that the endogenous analyte concentration would fall within the validated range of the assay[40].

The assay's repeatability was next assessed using five independently prepared samples analyzed on five different days. For each preparation, three aliquots of the representative sample matrix were spiked with high, medium, and low concentrations of heavy-labeled standard peptide (500x, 50x, and 2.5x the assay LLOQ, respectively). Total assay variability was determined at the transition level, and was required to be <20% at all three spiking concentrations[40]. Assays which did not meet these criteria were excluded (Fig. 3a and Supplementary Data 1). Across all developed assays, the average LLOQ was 18.15 fmol, which demonstrates the overall excellent sensitivity of these assays.

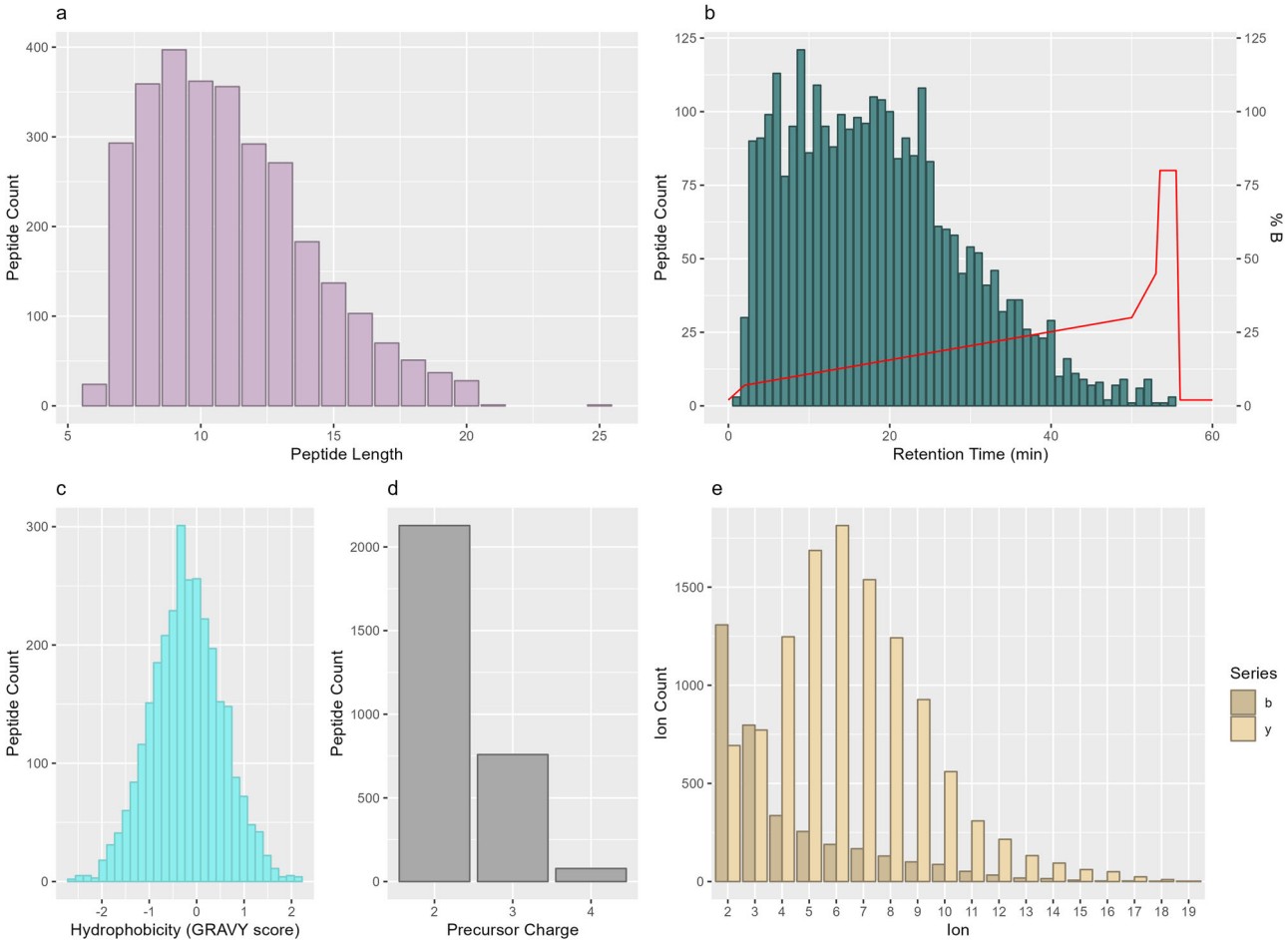

**Fig. 2 Peptide sequences selected for tissue-specific assay development.** The properties of the 2965 proteotypic surrogate peptides selected for MRM assay development. **a** peptide length **b** peptide retention time in minutes (teal bars) compared to percentage of organic buffer B (red line) in the gradient, where buffer B consists of 0.1% formic acid in acetonitrile **c** hydrophobicity as indicated by GRAVY score **d** precursor ion charge **e** identity of the top 5 most intense ions. Ion series type is shown by yellow (y ions) or tan (b ions).

In total, 7184 assays were successfully developed, corresponding to 2118 unique proteins (Fig. 3b and Supplementary Data 5). The number of assays developed in each organ or tissue ranges from approximately 100 (white and brown adipose tissue) to over 600 (brain and liver; Fig. 3b). Approximately 70% of assays were validated in more than one organ and tissue (Fig. 3c).

**Measurement of protein concentrations in target tissues.** Mouse models enable the study of the molecular mechanisms underlying health and disease, however, it is important to first identify sex- and strain-specific characteristics which may impact experimental results[55–58]. To demonstrate the power of our MRM-MS assays for molecular-level characterization and analysis, we measured normal protein concentrations in samples of 20 organs and tissues from six female and six male mice from three common background strains: C57BL/6NCrl, NOD/SCID, BALB/cAnCrl. To accomplish this efficiently, developed assays were multiplexed into organ- or tissue-specific panels, each containing approximately 125 proteins per panel. One to four panels were established per sample type and 31 unique panels were created in total (Supplementary Data 1). As long as the described acquisition method parameters are maintained (see Material and Methods for details), researchers can create customized panels within the organ or tissue of interest (e.g., brain assays from panels 5 and 6 can be combined). However, it is not possible to

use an assay validated in a one organ or tissue in a different sample type without additional validation.

Combining the results for all organs and tissues, 5149 concentrations were measured in total which corresponds to 1691 unique proteins. Of these, a small number (291 or 17% of all proteins assayed) were not quantified in any of the samples. This could have occurred for a number of reasons, including that the protein is below the assay's limit of detection, or is expressed only during certain biological states or as a result of disease. To aid biological interpretation, all protein abundances are summarized in the MouseQuaPro database (http://mousequapro.proteincentre.com)[54], which is linked to UniProt, DisGeNET, KEGG, and other knowledgebases. Data are searchable, for example, by protein and gene names, UniProt accession number, involvement in disease and pathway and drug associations[54].

Overall, unique patterns of protein expression were identified between each of the three strains in a number of organs and tissues (Figs. 4–6). We previously described the absence of immunoglobulin proteins from NOD/SCID mouse plasma[40]. Immunoglobulins are produced by B cells[59], and are therefore not present in NOD/SCID mice due to the impaired lymphocyte development in this strain stemming from the *Prkdc^scid* mutation. We show the absence of several immunoglobulins, including IGH-3 (Ig gamma-2B chain C region; P01867), IGHG1 (Ig gamma-1 chain C region secreted form; P01868), IGHM (Immunoglobulin heavy constant mu; P01872) and IGKC

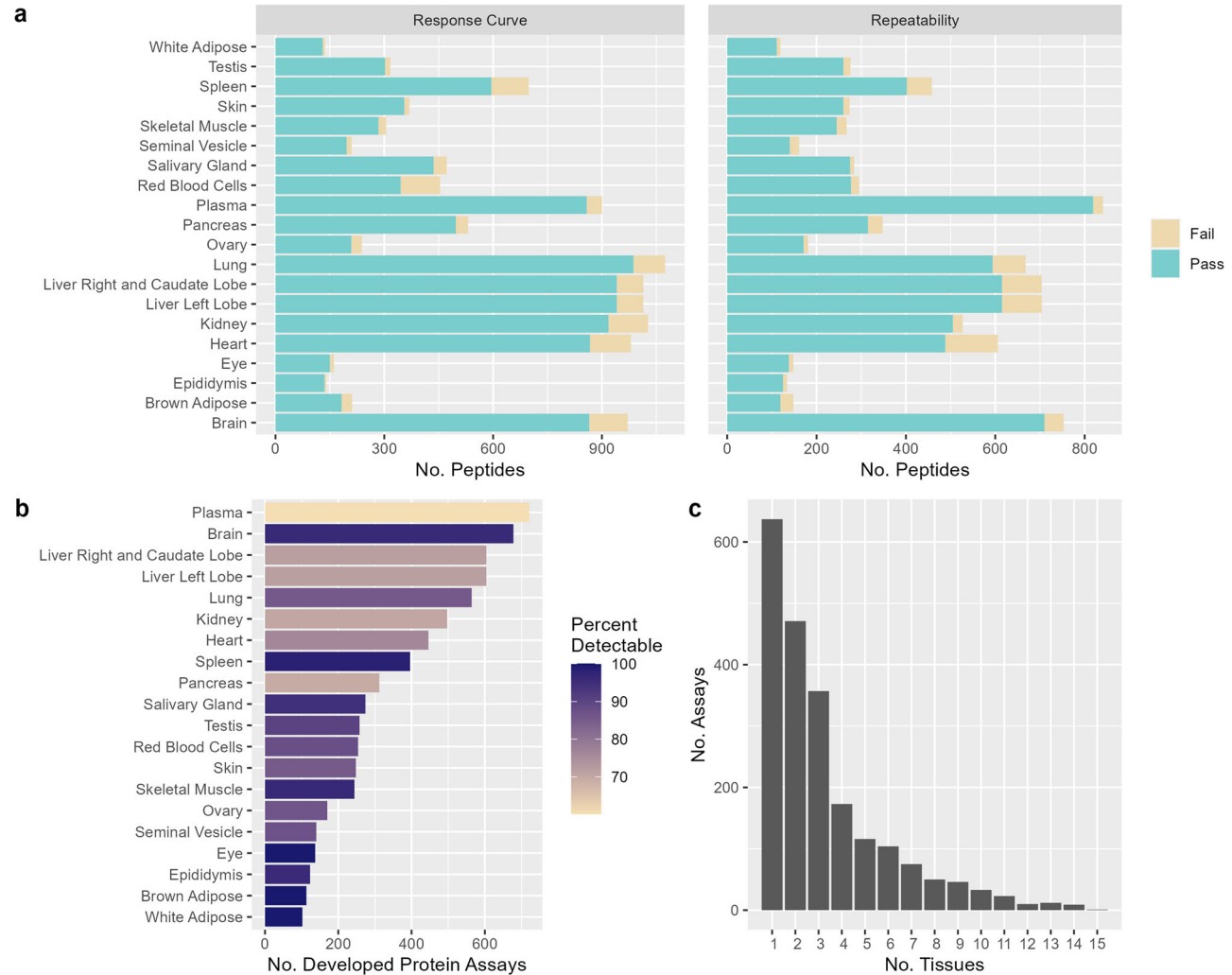

**Fig. 3 Summary of assay development in 20 mouse organs and tissues. a** shows the number of peptide assays for which a response curve was successfully characterized and assay variability was < 20% for each sample type. Number of passing assays are indicated in blue and failed assays by yellow **b** shows the number of unique proteins with developed assays in each organ or tissue, ranging from approximately 100 proteins in white adipose tissue to > 600 proteins in plasma and brain. Bars are shaded by the proportion of assays for which the endogenous protein was within the assay range in normal, healthy samples. The overlap of assays across sample types is shown in **c**, where 637 of the developed assays are unique to a single organ or tissue, and the remaining 1481 (70%) were validated in more than one sample type.

(Immunoglobulin kappa constant; P01837), across all tissues of this strain (Fig. 4a). The genetic diversity of immunoglobulin gene sequences across mouse strains has been documented[60,61] and may contribute to the expression patterns observed in BALB/cAnCrl and C57BL/6NCrl mice (Fig. 4a). Please note that to facilitate cross-tissue comparison, normalized concentrations are depicted. If desired, protein concentrations can be calculated from molecular weights of the surrogate peptides and the corresponding proteins, which are displayed in the MouseQuaPro database.

The extent of immune system disruption in NOD/SCID mice can be further observed in the spleen, which is a lymphoid organ rich in B and T lymphocytes. The four following immune-related proteins were significantly downregulated in the spleen of NOD/SCID mice ($p < 0.05$ determined by two-way ANOVA, adjusted for multiple testing using the Benjamini–Hochberg method; Fig. 4b): HAAO (3-hydroxyanthranilate 3,4-dioxygenase; Q78JT3) which is important for tryptophan catabolism, impacting T cell regulation and apoptosis[62–64]; LPXN (Leupaxin; Q99N69) and PYK2B (Protein-tyrosine kinase 2-beta; Q9QVP9) which are known interactors and immune regulators,

affecting B cells in the spleen[65–67]; and HMBG1 (High mobility group protein B1; P63158) which is a multifunctional protein that plays a role in V(D)J recombination, an essential process in lymphocyte development[68]. Interestingly, both PRKDC (which is mutated in NOD/SCID mice) and HMBG1 are associated with lymphoid-specific proteins RAG-1 and RAG-2 (Recombination-Activating Gene 1 and 2; P15919 and P21784), which cleave DNA and facilitate V(D)J recombination in both mice and humans[68–71]. Downregulation of HMBG1 in spleen may reflect the disruption of lymphocyte development: in other tissues of NOD/SCID mice, HMBG1 abundance was not affected (see MouseQuaPro for details), likely reflecting its other biological functions in these tissues.

Another example of strain-specific protein expression is the family of Alpha-1-antitrypsins (serpins), which includes up to six individual proteins in mice (Serpina1a, P07758; Serpina1b, P22599; Serpina1c, Q00896; Serpina1d, Q00897; Serpina1e, Q00898 and Serpina1f, Q9DCQ7). The specific combination of *serpina*-related genes varies according to the strain of the animal[72–74], which is reflected in our data (Fig. 5). Due to the high similarity of the Serpina1 proteins it was not possible to

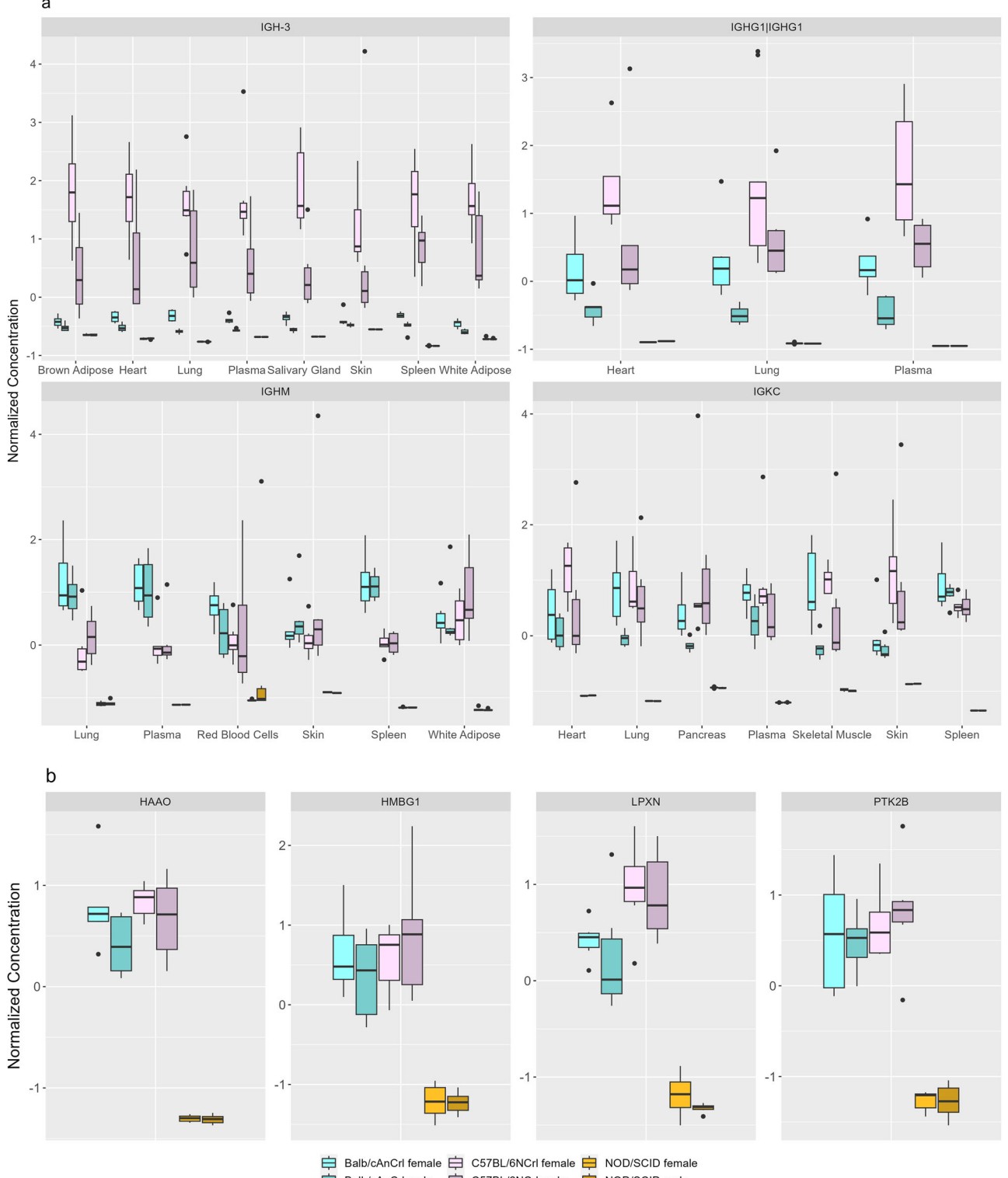

**Fig. 4 Concentrations of immune-related proteins in mouse organs and tissues.** Concentrations measured in samples from $n = 6$ male and $n = 6$ female mice from three strains. **a** shows the concentration of four immunoglobulin proteins in several mouse organs and tissues, displaying both strain- and sex-specific differences in protein concentration. **b** shows the concentration of four proteins associated with the regulation of lymphocytes. In the spleen, these proteins are deficient in NOD/SCID mice compared to the other strains. Normalized concentration values are shown.

select proteotypic peptides distinguishing Serpina1a from Serpina1c. Therefore, two assays measured the cumulative concentration of multiple serpins: Serpina1a-e (peptide sequence VINDFVEK; Fig. 5c) and Serpina1a/c (LAQIHFPR; Fig. 5d).

Three additional assays were specific to Serpina1b (LVQIHIPR; Fig. 5e), Serpina1d (ELISQFLLNR; Fig. 5a), and Serpina1e (LAQIHIPR; Fig. 5b). C57BL/6NCrl mice had the highest concentrations of Serpina1d and Serpina1e (Fig. 5b, c), while

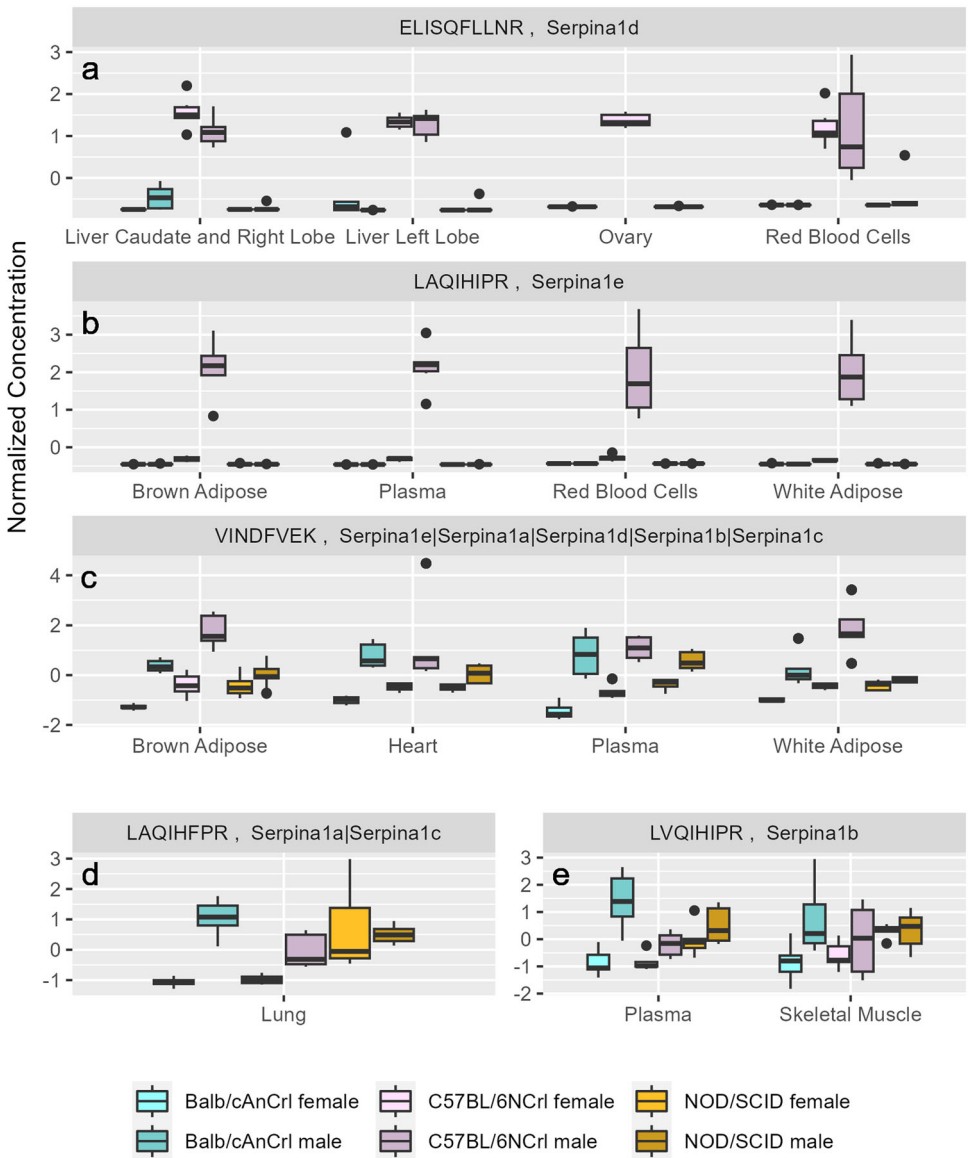

**Fig. 5 Concentration of serpin proteins in mouse organs and tissues.** Concentrations measured in samples from $n = 6$ male and $n = 6$ female mice from three strains. Mice have multiple serpina1 genes, which vary according to mouse strain. We used five unique assays **a–e** to measure cumulative and individual concentrations of Serpina1a-e proteins which demonstrate strain-specific expression patterns. Normalized concentration values are shown.

Serpina1b was present in all three mouse strains at similar levels (Fig. 5e). The concentration of some serpins also varied according to sex (Fig. 5).

The measured concentrations can also be combined to create specific proteomic profiles within each organ or tissue. To demonstrate this, spleen samples were distinguished by both strain and sex using principal component analysis (Fig. 6). Skeletal muscle, testis, epididymis, and plasma also showed strong separation based on strain, while salivary gland, kidney, and white adipose grouped primarily according to sex (Supplementary Fig. 1). Finally, some organs such as eye and skin showed little to no grouping according to the measured protein concentrations (Supplementary Fig. 1).

## Conclusions

Detailed molecular analysis of mouse organs and tissues is challenging due to the current lack of available tools for precise, robust, and accurate measurement of protein concentrations at a large-scale. MRM-MS is a highly reproducible and robust technique for protein quantitation[75]. Its well-defined transition lists, linear range, and strict validation criteria provide benefits such as shorter HPLC gradients and require less technical replicates than untargeted approaches. However, this method is also limited to developed assays, and therefore best suitable to answer specific biological questions that require accurate protein quantitation. To improve the availability of targeted assays and advance our knowledge of normal protein concentrations in three commonly used strains of laboratory mice, we developed 7184 quantitative MRM-MS assays corresponding to 2118 unique mouse proteins. Our assays measured proteotypic surrogate peptides for each target protein and underwent rigorous characterization and validation to ensure their selectivity, sensitivity, and robustness. Researchers can choose a subset of these assays for their specific purpose, or use this work as a guideline for development of their own MRM-MS assays.

Here, we showcased the use of MRM-MS for determination of protein abundances in 20 sample types from clinically healthy mice. In total, 5149 concentration measurements were obtained,

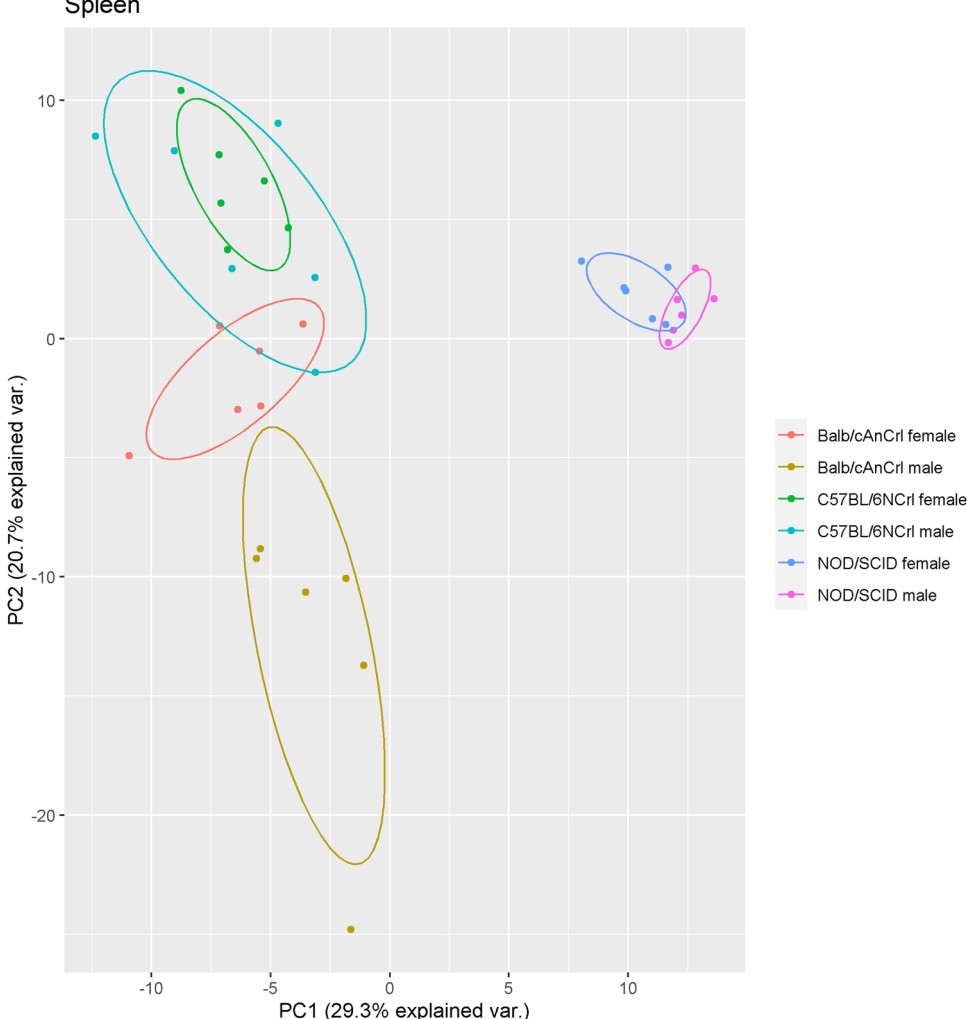

**Fig. 6 Principal component analysis plot of spleen samples from three mouse strains.** The first and second principal components account for 29.3% and 20.7% of the variability between spleen samples, respectively.

corresponding to 1691 unique proteins. Both sex- and strain-specific differences in protein concentration were identified in various organ and tissue types. The reference protein concentrations measured in various mouse organs and tissues are a valuable resource to any researcher using mouse models, and can be found in our MouseQuaPro database.

## Materials and methods

**Ethics statement.** All sample processing at the University of Victoria-Genome British Columbia Proteomics Center was performed under the approval granted by the University of Victoria Animal Care Committee. The Center for Phenogenomics (TCP; Toronto, ON, Canada) Animal Care Committee reviewed and approved all procedures conducted on animals at TCP and procedures were performed in compliance with the Animals for Research Act of Ontario and the Guidelines of the Canadian Council on Animal Care.

**Sample collection and processing.** Samples of 20 mouse organs and tissues were received from TCP including brain, eye, salivary gland, heart, lung, liver caudate and right lobe, liver left lobe, pancreas, spleen, kidney, ovary, testis, epididymis, seminal vesicle, skin, skeletal muscle tissue, brown and white adipose tissue, and blood separated into plasma and red blood cell portions. Samples were collected from 12-week-old mice from three strains, C57BL/

6NCrl, BALB/cAnCrl, and NOD/SCID, and immediately frozen in liquid nitrogen prior to shipment on dry ice. Samples were stored in −80 °C conditions at the University of Victoria-Genome British Columbia Proteomics Center until further processing. Except for plasma and red blood cell samples, which were digested directly, samples were homogenized by lyophilisation followed by bead-beating[76]. Briefly, the frozen sample was weighed and lyophilized overnight. The dried material was then combined with 3 × 3.2 mm stainless steel beads in a micro-centrifuge tube and homogenized by bead-beating on the MM 400 homogenizer (Retsch, Haan, Germany). Shaking was performed using 3 × 1 min intervals at 27 Hz, with 1 min rest periods on ice between each cycle, until samples were completely ground. Samples were subsequently rehydrated with buffer (4.5 M Urea, 200 mM Tris) at a ratio of 1:4 (w/v) prior to cleanup by acetone precipitation. The resulting protein pellets were rehydrated by adding 9 M Urea, 300 mM Tris, pH 8.0 at one half to two times the initial homogenate volume.

**Tryptic digestion.** Protein content of each homogenized sample was measured by bicinchoninic acid assay (BCA; Thermo Fisher Scientific, Ottawa, ON, Canada) or Bradford assay (Sigma-Aldrich, St. Louis, MO, USA) according to the manufacturer's instructions. Samples were transferred to a 96 well plate (Eppendorf, Mississauga, ON, Canada) for digestion using a

Tecan Evo (Männedorf, Switzerland) liquid handling robot. Samples used for assay development were pooled prior to digestion; samples from individual mice for concentration measurement were added to the 96 well plate in a randomized order. A control was created for each organ or tissue type by pooling a small volume from each individual mouse and was distributed throughout the plate. Bovine serum albumin (BSA; Sigma Aldrich, Oakville, ON, Canada) solution (1 µg/µL in sample buffer) was added in the last wells of the plate and processed in parallel with tissue samples.

Tryptic digestion was performed by denaturing proteins with 20 mM dithiothreitol in 9 M urea, 300 mM Tris, pH 8.0 and incubating samples at 37 °C for 30 min. Next, samples were alkylated in 40 mM iodoacetamide at room temperature for 30 min, and subsequently diluted 10-fold in 100 mM Tris pH 8.0. Trypsin (Worthington Biochemical Corporation, Lakewood, NJ, USA) was added at a protein:enzyme ratio of 10:1, and samples were digested for 18 h at 37 °C while shaking at 500 rpm. Digests were acidified to 1% (v/v) formic acid and spiked with synthetic peptide mixtures prior to desalting and concentration by solid phase extraction using OASIS HLB cartridges (Waters, Milford, MA, USA) according to manufacturer's instructions. Eluted samples were lyophilized to dryness and re-suspended in 0.1% formic acid. Twenty micrograms of each sample were injected on the HPLC column for each MRM experiment, or 1.5 µg sample for untargeted MS analysis.

**Untargeted MS analysis.** Samples of 20 organs and tissues from $n = 3$ male and $n = 3$ female C57BL/6NCrl mice were pooled according to sex and analyzed by untargeted analysis on an Orbitrap Fusion Tribrid instrument coupled to an EASYnLC 1000 liquid chromatography system via a Nanospray Flex NG source (Thermo Fisher Scientific, Waltham, MA, USA). Digests were injected onto a reversed-phase pre-column (100 µm internal diameter, 2 cm length, using Magic C18-AQ 5 µm particles, 100 Å pore size) followed by a reversed-phase nano-analytical column (75 µm internal diameter, 15 cm length, 5 µm particles, 100 Å pores) (Michrom BioResource, Auburn, CA, USA). The solvents used for the HPLC gradient were 2% acetonitrile, 0.1% formic acid (A) and 90% acetonitrile, 0.1% formic acid (B). Separation was performed using a 140 min gradient at a flow rate of 300 nL/min, as follows (%B, time in min): 3, 0; 35, 110; 45, 120; 100, 130; 100, 140. The analytical column was coupled to a 10 µm emitter (New Objective, Woburn, MA, USA) and acquisition was performed using 2500 V spray voltage and 275 °C capillary temperature. Data-dependent acquisition collected precursor spectra (400–1800 m/z) in the Orbitrap analyzer at a resolution of 120,000 and an automatic gain control target of 400,000. Fragment spectra were collected in the ion trap with an automatic gain control target of 10,000 and maximum injection time of 35 ms. The isolation window was set to 1.6 Da and higher-energy collisional dissociation was used for fragmentation with a stepped collision energy of 35% ±5. Dynamic exclusion was set to exclude ions for 10 s using a Δ of 10 ppm, after selecting 2 times within 5 s. Acquired raw data from males and females of each sample type were processed together in Proteome Discoverer (version 2.2.0.388) which searched the data against the mouse reference proteome (UniProt) using the MASCOT search engine. The search settings were 6 ppm precursor mass tolerance, 0.6 Da fragment mass tolerance, maximum 1 missed cleavage, deamidation (N, Q) and oxidation (M) as variable modifications and carbamidomethyl (C) as a fixed modification. Results were filtered to a 1% false discovery rate, high confidence peptides only, and a minimum of two peptides per protein.

**Selection of protein targets.** The initial selection criteria involved searching the UniProt database (https://www.uniprot.org/) with keywords relevant to the target organ or tissue (morphology features, function, and known pathologies) and cross-referencing the resulting list with the list of proteins identified by discovery proteomics. Next, if the endogenous peptide was readily detectable by MRM-MS in the tissue of interest, we continued with assay development. However, if the endogenous peptide was not readily detectable by MRM-MS, it was only kept for further assay development if a disease association was described in the literature (arguing that there might be a potential upregulation of protein expression in a disease model). We also developed MRM-MS assays for highly abundant proteins in each tissue.

**Synthetic peptide standards.** Peptide standards were synthesized as previously described[77]. Stable isotope-labeled peptides were synthesized by coupling $^{13}C/^{15}N$ N-Fmoc L-arginine and L-lysine (98% isotopic enrichment; Cambridge Isotope Laboratories, Andover, MA, USA) to TentaGel® R TRT-Cl resin (RAPP Polymere, Tübingen, Germany). Synthesis was performed in dimethylformamide with a 10x or 20x amino acid excess, using 40% piperidine for Fmoc deprotection, and HCTU (1 eq)/NMM (2 eq) as activator/base reagents. Peptides were subsequently cleaved from the resin, purified through reversed-phased HPLC fractionation, and characterized in-house by amino acid analysis and capillary zone electrophoresis. Synthesis of unlabeled standard peptides followed the same procedure, except using Wang resins preloaded with non-modified N-Fmoc lysine and arginine (Matrix Innovations, Quebec City, QC, Canada).

**MRM-MS analysis.** MRM-MS was performed using an Agilent 6490/6495 series Triple Quadrupole coupled to an Agilent 1290 Infinity UHPLC system (G4220A). Twenty micrograms of each digest were injected onto an Agilent Zorbax Eclipse Plus C18 Rapid Resolution HD column (2.1 × 150 mm, 1.8 µm particles) maintained at 50 °C. The solvents used for the HPLC gradient were 0.1% formic acid in water (A) and 0.1% formic acid in acetonitrile (B), and the following gradient was used at a flow rate of 0.4 mL/min: (%B, time in min): 2, 0; 7, 2; 30, 50; 45, 53; 80, 53.5; 80, 55.5; 2, 56 with a 4 min post-gradient equilibration at 2% B. Transitions were monitored in positive ion mode using dynamic MRM acquisition with a detection window of 1 min, < 900 ms cycle time, and a dwell time of at least 9 ms. All MRM raw data were processed and inspected using the Skyline Daily software.

**Development of quantitative MRM-MS assays.** Each MRM-MS assay consists of a stable isotope-labeled synthetic peptide standard ("heavy") used for normalization, and an unlabeled synthetic peptide standard ("light") used for generation of the calibration curve. Protein concentration is established based on the measured abundance of the surrogate tryptic peptide originating from the tissue (endogenous; also "light"), normalized to the heavy-labeled standard. The ratio of endogenous:heavy is then read off the calibration curve to calculate the concentration. Note that for the assay development experiments, the roles are reversed: the combined light signal (consisting of the endogenous plus synthetic "light" peptide) is used as the normalizer and heavy peptide is used for the response curve. Further details on the design of validation experiments have been published previously[40].

*Selection of transitions.* First, chromatographic retention times were determined using the synthetic peptide standards. The optimal precursor ion charge state, collision energy, and fragment

ions were then empirically determined for each peptide. These parameters were applied to all subsequent experiments. The five best-responding transitions were monitored by MRM-MS for each peptide in all subsequent assay development experiments.

*Spiking peptide standards for assay development.* Pooled matrix samples were prepared for each organ and tissue by combining homogenized and digested samples from 3 male and 3 female C57BL/6NCrl mice. Please note that for assay development, liver was not separated into the individual lobes. For each assay, the heavy-labeled peptide standard was spiked into the pooled matrix and the endogenous peptide was used as the normalizer. If the concentration of an endogenous peptide was too low or it was not detectable, a corresponding unlabeled synthetic peptide was added to boost the light signal (200 fmol per injection). The peak area of the heavy signal was then normalized to the peak area of the light signal (endogenous only or endogenous plus unlabeled synthetic peptide).

*Response curve.* A twelve-point dilution series using heavy-labeled peptide standards (20,000 to 0.3125 fmol peptide/injection; dilution pattern: 1:10:10:2.5:2:2:2:2:2:2:2:2) was spiked into pooled matrix sample. Technical triplicates of the individual samples were injected in order of lowest to highest concentration. The peak area ratios were plotted against the known concentration of spiked heavy-labeled peptides at each concentration level and analyzed using a $1/x^2$ weighting to determine the assay's linear range and LLOQ. At least three concentration points on the curve with a coefficient of variation <20% and accuracy ±20% of the expected concentration were required for an assay to pass[40]. Assays that did not meet these criteria are marked "fail" in Supplementary Data 1.

*Repeatability.* On five separate days, five independent aliquots of pooled matrix sample were spiked with heavy-labeled peptide standard at 500x, 50x, and 2.5x of the assay LLOQ determined in the response curve experiment. Samples were subsequently analyzed by MRM-MS on five different days. Each spiked aliquot was injected in triplicate for a total of 45 measurements per target peptide. Three criteria were used to fail or pass the assay at this step, and were evaluated per each transition monitored: intra-assay variability, inter-assay variability, and total assay variability. The intra-assay variability was defined as the average coefficient of variation at each spiking concentration. The inter-assay variability was defined as the average coefficient of variation of each injection over all five days. Finally, the total assay variability was determined at each spiking concentration as the root sum of the squares of the intra-assay variability and inter-assay variability[41]. Total assay variability was required to be less than or equal to 20% at all concentration points, for at least one transition, for an assay to pass this stage of development[40].

An example of assay development (response curve and variability data) is provided in the Supplementary Information (Supplementary Figs. 2–4).

**Protein quantitation.** For the measurement of protein abundances in mouse organs and tissues, the MRM-MS assays were grouped into sample-specific panels to ensure that acquisition criteria (cycle time: <900 ms, dwell time: >9 ms, and retention time window: 1 min) were met. Therefore, each panel consisted of approximately 125 peptide targets and the three best-responding fragment ions were monitored per peptide. Depending on the number of assays developed per organ or tissue, each sample type was measured using one to four MS experiments, corresponding to measurements with one to four individual panels

(Supplementary Data 1). Endogenous peptides were quantified in samples from individual female ($n = 6$) and male ($n = 6$) mice using an 8-point external calibration curve, prepared as previously described[40]. Briefly, the calibration curve was constructed by spiking synthetic light peptides (ranging in concentration from 1 to 1000x assay LLOQ) into digested BSA as a surrogate matrix. Heavy-labeled peptides were added to all samples and standards at 100x LLOQ as the normalizer. Sample protein concentrations were calculated based on the calibration curve analyzed using a $1/x^2$ weighting. The concentration data for each organ or tissue type was filtered to remove "undetectable" peptides for which 50% or more of all measurements were below one half the assay's LLOQ.

**Statistics and reproducibility.** Pooled matrix samples for the development of quantitative MRM-MS assays were prepared by combining homogenized, digested organ or tissue samples from $n = 3$ male and $n = 3$ female C57BL/6NCrl mice. Protein quantitation experiments measured individual organ or tissue samples from $n = 6$ male and $n = 6$ female C57BL/6NCrl, BALB/cAnCrl, and NOD/SCID mice (36 mice total). Peptides which had a significantly different concentration between groups were determined by two-way ANOVA, and $p$-values were adjusted for multiple testing using the Benjamini–Hochberg method. Group means were compared by Tukey HSD test. Comparisons were considered significant if $p < 0.05$. For cross-tissue comparison of protein expression (Figs. 4 and 5), protein concentrations were z-score normalized for each organ or tissue type prior to plotting.

**Reporting summary.** Further information on research design is available in the Nature Portfolio Reporting Summary linked to this article.

## Data availability

The raw data from the untargeted MS experiments have been deposited to the ProteomeXchange Consortium via the PRIDE partner repository with the dataset identifier PXD021333. A summary of the peptides and proteins identified by untargeted MS is available in Supplementary Data 2 and 3; data generated during assay development are found in Supplementary Data 4 and 5. The reference sample concentration data can be accessed via Panorama Public at https://panoramaweb.org/MRMmouse20tissues.url and at ProteomeXchange with ID number PXD020930. The protein abundances measured in tissues from three commonly used mouse strains during this study are also hosted in an online, interactive knowledgebase, MouseQuaPro[54], which displays the concentration of each protein along with additional information about the protein's biological function, human orthologues, and involvement in disease.

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

## Acknowledgements

The University of Victoria-Genome British Columbia Proteomics Center is grateful to The Center for Phenogenomic's Pathology Core for their support and assistance, and to Genome Canada and Genome British Columbia for financial support through the Genomics Innovation Network (project codes 204PRO for operations and 214PRO for technology development), the Genomics Technology Platform (GTP - project code 264PRO), the Bioinformatics and Computational Biology program (project code 282PQP), and the Disruptive Innovation in Genomics program (DIG – project code 234DMP) for the development of the mouse assays. C.H.B. is grateful for support from the Genomics Technology Platform (GTP - project code 264PRO) and the Segal McGill Chair in Molecular Oncology at McGill University (Montreal, QC, Canada). C.H.B. is also grateful for support from the Warren Y. Soper Charitable Trust and the Alvin Segal Family Foundation to the Jewish General Hospital (Montreal, QC, Canada). T.C.P. acknowledges the support of Genome Canada and Ontario Genomics (GTP, OGI-137). This work was done under the auspices of a Memorandum of Understanding between the University of Victoria, McGill University, and the U.S. National Cancer Institute's International Cancer Proteogenome Consortium (ICPC). ICPC encourages international cooperation among institutions and nations in proteogenomic cancer research in which proteogenomic datasets are made available to the public. This work was also done in collaboration with the U.S. National Cancer Institute's Clinical Proteomic Tumor Analysis Consortium (CPTAC).

## Author contributions

Study conception: C.H.B. and A.S.; Experiment design: S.A.M., David Schibli, Derek Smith, and Y.M.; Contributed the mouse tissue samples: C.M., L.M.J.N., A.M.F., and M.G.; Performed and analyzed the experiments: S.A.M., N.J.S., H.P, A.L.K., A.M.J., J.C.M., D.B.H.; Software design: P.B. and Y.M., Provided financial support: C.H.B. and A.S.; Wrote the manuscript: S.A.M. and H.P.; all authors read and contributed to the final version of the manuscript.

## Competing interests

The authors declare the following competing interests: C.H.B. is the co-founder and CSO of MRM Proteomics, Inc. and the VP of Proteomics at Molecular You. The other authors declare no competing interests.
