## [Peer Review File · Communications Biology]

Reviewers' comments:

Reviewer #1 (Remarks to the Author):

The authors report on the development of large-scale multiple reaction monitoring (MRM) assays for large-scale quantitation of proteins from 20 mouse organs and tissues. The experiments were well designed, and the data is solid and have shown that the MRM assays can be used for quantitative molecular phenotyping. However, there are several major concerns for this manuscript:

1. In the section of Introduction, it is not clear why the largest MRM assays are important to the scientific community. When compared to untargeted proteomics (i.e., global proteomics) the development and implementation of large-scale MRM assays are too expensive (e.g., the assay development time, the cost for synthesizing of 1000s of pure isotope-labeled heavy peptide standards with \$300-400 per one pure standard, and LC-MS running time). In the discovery phase using MRM assays is cumbersome and costly. More explanations and even the cost comparison between global and targeted proteomics are required for readers to understand when they need use the large-scale MRM assays.
2. In pages 9-10 the authors mentioned the identification and selection of proteins and peptides for MRM assay development. But still, the selection criteria are not clear (e.g., biological significance of the selected proteins). Most proteins were selected based on the detectability from untargeted proteomics and some proteins not identified by untargeted proteomics were also selected. Supplementary table with the detailed information of why these proteins were selected is needed, which could help readers to select the MRM assays of interest for their projects.
3. In the section of Materials and Methods, the authors did not mention that how many LC-MRM measurements had been run to screen all the selected target proteins for complete analysis of one single sample. Based on the Skyline files provided by the authors, many runs were performed iteratively for one sample. Detailed information (e.g., the retention time scheduling and the number of LC-MRM runs for each sample) is required.

Reviewer #2 (Remarks to the Author):

The manuscript entitled "Multiple reaction monitoring assay for large-scale quantification of proteins from 20 mouse organs and tissues" describes the development of multiple reaction monitoring MS assays in mice. The authors used state-of-the-art MS and synthesized a large number of isotopically labelled peptides and quantified selected candidates in three different mouse models. The approach enabled the quantification of 1691 proteins among all tested tissues.

Overall, the development of MRM assays is a versatile tool to accurately quantify protein changes in complex biological samples. Unfortunately, the selection of immunoglobulins in Figure 4 demonstrates the weakness of the study. Those proteins are quite abundant and they are usually detectable with untargeted approaches. Moreover, it is not clear whether immunoglobulins as shown in Figure 4 are blood contaminations or reflect the uptake into the tissue. In addition, the authors showed in previous publication the absence of these factors in NOD/SCID mutants, which is not surprising. Similar, the quantification of the selected serpins might be also detectable by untargeted approaches. A more tissue and cell type-specific selection of proteins would be more helpful compared to the list of abundant proteins.

It would be helpful to compare MRM quantification with non-targeted approaches in order to highlight the more accurate quantification, if that is the case. Why have the authors not selected more functional relevant proteins, including tissue-specific proteins, signaling molecules or the bridging integrator 2 as described on page 10? In addition, the development of the MRM assays is also not very well documented and difficult to understand for non-proteomics experts. It would be helpful to plot some of the curves for the determination of the LLOQ value, at least in the supplement. The selected peptide/protein quantifications should be statistically compared with untargeted approaches. Would it be possible to calculate absolute protein concentrations/copy numbers of the hits?

The selection of 2676 seems rather arbitrary and it is not clear which proteins were selected from the initial untargeted analysis? Which candidates were identified or selected by a relevant association to a disease such as the bridging factor 2. The authors selected 2676 proteins (page 10, line 290) for assay development and synthesized 2965 peptides. What is the average number of peptides used for MRM assay per protein? Overall, the authors developed 2118 assays. What happened with the remaining ~500 proteins?

A weakness of the study is the initial number of identified proteins per tissue. So far, several studies have described protein expression in mouse tissues, and here the numbers based on untargeted MS are higher compared to the present study (Figure 1). For example, skeletal muscle tissue has been measured with ~500 proteins and in tissues such as spleen and skin ~1000 proteins have been identified. Even with in-solution digestion and the high-resolution fusion device used, more protein hits could be expected. The authors cited the Geiger publication. However, this work was from "2013" and used in-vivo SILAC, which results usually in less proteins hits due to the LysC digest compared to trypsin digestion. With a modern state-of-the-art MS one should expect definitely more hits. Finally, the Figure 3B summarized the number of assays and the skeletal muscle as well as the skin and testis reflect ~220 assays. It remains questionable whether this limited size of targets reflects a useful resource. Moreover, the authors showed only Uniprot identifier in the SI data? Why is there no protein/gene name? That would be very helpful for users to find their target of interest. In addition, the intensity and number of peptides in the untargeted analysis might be helpful data in combination with the MRM transitions.

SI table 1 is SI table 3. Table 2 shows the peptides hits of the untargeted analysis. It seems there is mixing of SI tables. SI table 3 is SI table 2. Moreover, the tables are far from a "detailed" summary. Why is it not possible to mention the protein names in the table? This is very inconvenient for the non-expert. The intensities of peptides, scores etc. are missing to better judge the quality of the data. The number of peptides per protein is also not listed. The summary of the protein group might be also helpful. Since muscle and heart show a high number of isoforms it would be nice to see whether all selected peptides are unique.

Reviewer #3 (Remarks to the Author):

Comments

#1 The study by Michaud et al provides a large scale of assays to MRM in 20 organs in multiple mouse strains. The quantification of each peptide has also been confirmed based on the CPTAC guidelines, which is a reliable result. Broad range of applications such as examination of biomarkers and proof of drug mechanism of action are expected.

#2 The authors have added assays for undetected proteins based on biological importance. (Line 285). The list of added proteins should also be shown separately from the set of detected proteins.

#3 Figure 3 shows the number of peptide assays evaluated in each organ, but the numbers are quite different from the number of proteins identified for each organ shown in Figure 1. For example,

plasma and erythrocytes are significantly increased compared to proteins identified from organs, but white fat, brown fat, epidymis, ovary, and testis appear to be decreased.

Is the increase in proteins due to the added biological significant proteins mentioned on line 285? For organs where the number of assays is reduced relative to the number of proteins identified, it would be preferable to state the reason for this, such as whether they are judged to be of low biological importance or because there are many peptides that are not suitable for quantification.

#4 The authors develop assays in panels so that a total of 31 panels were designed with 1-4 panels for each organ sample. Each panel contains about 125 proteins (Line 355-356). The authors should display the list of proteins covered by each panel and indicate which panel was used for which organ.

#5 Line 409, 'The concentration of some SERPINA1a proteins further', but this sentence is difficult to understand. Should it be 'Serpin proteins' rather than 'SERPINA1a proteins'?

#6 The authors should give the full names for RAG1 and RAG2 (Line 396).

We thank all the reviewers for their comments that allowed us to improve our manuscript. One of the main concerns was the biological relevance of our assays. To showcase how our MRM-MS assays can aid the life science community, the MouseQuaPro database was created (<http://mousequapro.proteincentre.com/>¹). Assay information for the proteins of interest can be found by the protein or gene names, involvement in diseases and pathways, or drug associations. The database also contains the protein abundances measured from three commonly used laboratory mouse strains, C57BL/6NCrI, BALB/cAnNCrI, and NOD/SCID, across 20 distinct organs and tissues. The MRM-MS assays described in this manuscript, together with the measured protein abundances hosted in MouseQuaPro database, provide a valuable tool to aid research involving mouse models as follows: **(a) guiding the experimental design.** For example, researchers can search which proteins are present in high abundance in the tissue of interest, or if sex should be considered as a biological variable in their specific experiment. This knowledge is a vital resource that can save time and cost in pilot studies. **(b) aiding accurate protein quantification.** Researchers can select a single MRM-MS assay or create a customized panel to interrogate their tissue of interest. We have already done all the time and cost-consuming optimization and validation, and these assays are ready-to-use once the respective standards are purchased. The cost of standards is comparable with the cost of an antibody. Being able to perform these assays does not necessarily involve having a mass spectrometer available in one's laboratory; triple quadrupole (QQQ) instruments required for these assays are the workhorses in most core mass spectrometry facilities. We modified the abstract (**p 2, lines 39-48**), introduction (**p 4, lines 94-98**) and conclusions (**p 21, lines 520-537**) accordingly. Please find our answers to the individual comments below; corresponding changes in the manuscript itself are highlighted in yellow.

Reviewer #1 (Remarks to the Author):

The authors report on the development of large-scale multiple reaction monitoring (MRM) assays for large-scale quantitation of proteins from 20 mouse organs and tissues. The experiments were well designed, and the data is solid and have shown that the MRM assays can be used for quantitative molecular phenotyping. However, there are several major concerns for this manuscript:

1. In the section of Introduction, it is not clear why the largest MRM assays are important to the scientific community. When compared to untargeted proteomics (i.e., global

proteomics) the development and implementation of large-scale MRM assays are too expensive (e.g., the assay development time, the cost for synthesizing of 1000s of pure isotope-labeled heavy peptide standards with \$300-400 per one pure standard, and LC-MS running time). In the discovery phase using MRM assays is cumbersome and costly. More explanations and even the cost comparison between global and targeted proteomics are required for readers to understand when they need use the large-scale MRM assays.

We agree with the reviewer's suggestion to expand the justification for the use of MRM-MS assays. We did not intend to suggest the use of MRM-MS in place of discovery experiments. MRM-MS assays represent an alternative to the traditional semi-quantitative techniques, such as Western blot or ELISA, and are especially important for protein targets with no available antibodies. As such, MRM-MS should be utilized when precise quantification is required (*e.g.*, a researcher is studying changes in expression of a specific group of proteins). In contrast to Western blot or ELISA, the MRM-MS assays provide exact measurements of abundances and can be easily multiplexed. The price of antibodies is comparable to that of synthetic peptide standards. Many researchers wishing to implement MRM-MS in their experiment will use only a subset of these assays (*e.g.*, associated with a specific tissue, pathway or disease). We clarified the intended use of MRM-MS assays in the introduction (**p 3, lines 77-85 and p 4, lines 94-98**), results (**p 17, lines 435-441**), and conclusions (**p 21, lines 520-537**).

The cost of developing MRM-MS assays was one of the motivating factors for this research. We dramatically reduced the time and cost required to implement MRM-MS in other laboratories by providing access to our optimized and fully validated assays (including the selection of appropriate surrogate peptides, determination of interference-free transitions, lower limit of quantitation, linear range, and repeatability) (**p 2, lines 41-43**).

Finally, via our new MouseQuaPro database (<http://mousequapro.proteincentre.com/>¹), all measured protein abundances in samples from three commonly used mouse strains are provided. The research community can use these values for hypothesis formulation and experimental design (**p 2, lines 43-48**). Data obtained by our assays are biologically relevant, as they highlight sex- and strain-specific differences in protein abundances across all 20 organs and tissues.

2. In pages 9-10 the authors mentioned the identification and selection of proteins and peptides for MRM assay development. But still, the selection criteria are not clear (e.g., biological significance of the selected proteins). Most proteins were selected based on the

detectability from untargeted proteomics and some proteins not identified by untargeted proteomics were also selected. Supplementary table with the detailed information of why these proteins were selected is needed, which could help readers to select the MRM assays of interest for their projects.

We modified the Material and Methods section to clarify selection of target proteins (**p7, lines 196-205**).

We cannot decide if the selected proteins are relevant or not, as hundreds of laboratories utilizing mouse models will have hundreds of different research interests with targets that might change in the context of new data². We would therefore like to refer the reviewer and the reader to our MouseQuaPro database (<http://mousequapro.proteincentre.com/>¹) where the assay information and measured protein abundances can be found for 20 tissues and organs across three laboratory mouse strains (male and female). Assays can be selected by protein and gene names as well as involvement in disease and pathways or drug associations. MouseQuaPro is directly linked to UniProt, DisGeNET, KEGG and other knowledgebases, and therefore updated as new information becomes available. We added references to MouseQuaPro throughout the manuscript (**p 2, lines 46-48, p 11, lines 313-317, p 13, lines 363-365 and p 17, lines 446-450**).

3. In the section of Materials and Methods, the authors did not mention that how many LC-MRM measurements had been run to screen all the selected target proteins for complete analysis of one single sample. Based on the Skyline files provided by the authors, many runs were performed iteratively for one sample. Detailed information (e.g., the retention time scheduling and the number of LC-MRM runs for each sample) is required.

For the measurement of protein abundances in murine tissues, the MRM assays were grouped into tissue-specific panels to ensure that MRM-MS method criteria (cycle time: < 900 ms, dwell time: > 9 ms, and retention time window: 1 min) were met. Therefore, each panel consisted of approximately 125 peptide targets and three best-responding fragment ions were monitored per peptide. Depending on the number of assays developed per tissue, one tissue sample was measured using one to four MS experiments, corresponding to measurements with one to four individual panels. We changed the Material and Methods and Results sections accordingly and added the information on panel numbers in the supplementary material (**p 10, lines 281-287, p 17, lines 435-441 and Supplementary Table 1**).

Reviewer #2 (Remarks to the Author):

The manuscript entitled “Multiple reaction monitoring assay for large-scale quantification of proteins from 20 mouse organs and tissues” describes the development of multiple reaction monitoring MS assays in mice. The authors used state-of-the-art MS and synthesized a large number of isotopically labelled peptides and quantified selected candidates in three different mouse models. The approach enabled the quantification of 1691 proteins among all tested tissues.

1. Overall, the development of MRM assays is a versatile tool to accurately quantify protein changes in complex biological samples. Unfortunately, the selection of immunoglobulins in Figure 4 demonstrates the weakness of the study. Those proteins are quite abundant and they are usually detectable with untargeted approaches. Moreover, it is not clear whether immunoglobulins as shown in Figure 4 are blood contaminations or reflect the uptake into the tissue. In addition, the authors showed in previous publication the absence of these factors in NOD/SCID mutants, which is not surprising. Similar, the quantification of the selected serpins might be also detectable by untargeted approaches. A more tissue and cell type-specific selection of proteins would be more helpful compared to the list of abundant proteins.

Since no enrichment strategies were involved in the sample preparation, it is expected that more abundant proteins will be detected preferentially. The strength of our approach is in precise quantification of protein abundance, which cannot be achieved without the use of isotopically labeled standards and the thorough assay validation described in the paper. We would like to refer the reviewer and the reader to the MouseQuaPro database (<http://mousequapro.proteincentre.com/>¹) where customized plots of protein abundance across sample types and mouse strains can be easily generated. We added references to MouseQuaPro throughout the manuscript (p 2, lines 46-48, p 11, lines 313-317, p 13, lines 363-365 and p 17, lines 446-450).

We revised Figure 4 to contain more immune-related proteins (p 18; below) and revised the corresponding section of the manuscript (p 17-19, lines 451-486). We also removed the statistical comparison of Ig levels between the C57BL/6NCrl and BALB/cAnCrl mice.

Figure 1. Concentrations of immune-related proteins in mouse organs and tissues. Panel A shows the concentration of four immunoglobulin proteins in several mouse organs and tissues, displaying both strain- and sex-specific differences in protein concentration. Panel B shows the concentration of four proteins associated with the regulation of lymphocytes. In the spleen, these proteins are deficient in NOD/SCID mice compared to the other strains. Normalized concentration values are shown.

2. It would be helpful to compare MRM quantification with non-targeted approaches in order to highlight the more accurate quantification, if that is the case. Why have the authors not selected more functional relevant proteins, including tissue-specific proteins, signaling molecules or the bridging integrator 2 as described on page 10?

Inherently, certain biologically interesting proteins (*e.g.*, cytokines and signalling molecules), cannot be detected or quantified by mass spectrometry due to their very low abundance. Enrichment and fractionation strategies might exist for some of these proteins, however specialized sample preparation prior to MRM-MS is not the focus of this study. We added a paragraph on MRM-MS limitations (**p 21, lines 520-524**). Furthermore, as indicated in our response to Reviewer #1 (point #2), it is not our intention to decide for researchers which proteins are biologically relevant for their research².

In addition, the development of the MRM assays is also not very well documented and difficult to understand for non-proteomics experts.

We have expanded the Material and Methods and Results section on assay development to clarify the process (**p 8-10, lines 231-278 and p 14-15, lines 383-411**).

It would be helpful to plot some of the curves for the determination of the LLOQ value, at least in the supplement.

An example response curve is now included as supplementary material (Supplementary Figure 2) as well as example repeatability data (Supplementary Figures 3 and 4). We also clarified the Response curve section in Material and Methods (**p 9, lines 255-263**).

The selected peptide/protein quantifications should be statistically compared with untargeted approaches.

We thank the reviewer for this suggestion. However, it is not clear what would be the outcome of such analysis due to the inherent differences between these two techniques, particularly the use of stable isotope-labelled peptide standards. In addition, the MRM-MS and untargeted analysis provide results in units that are not compatible (MRM-MS is concentration based, while untargeted analysis provides relative abundances if standards are not used). The closest comparison found was recently made by Williams *et al*³ who determined no statistically significant difference in protein abundances obtained by parallel reaction monitoring (PRM; an equivalent of MRM on an Orbitrap instrument) and data dependent acquisition (DDA). However, PRM and DDA can be performed on a single instrument, which is not the case of MRM-MS and untargeted analysis.

Furthermore, Williams *et al* focused on a small number of proteins only and used stable isotope-labelled standards in both workflows, simplifying data normalization and comparison between the techniques.

We would like to emphasize that we are not suggesting that MRM-MS should replace untargeted analysis. They both have their place, depending on the research question and the experimental design. For example, if a group of proteins should be measured repeatedly across a large number of samples, MRM-MS is the method of choice. Its well-defined transition lists, linear range and strict criteria for quantification allow for shorter LC gradients and less technical replicates. On the other hand, if a large number of undefined proteins should be quantified in a small number of samples, researchers might be better off using untargeted proteomics. To clarify the intended use of our assays, we modified the manuscript where appropriate (**p 3, lines 77-85, p 17, lines 435-441 and p 21, lines 520-531**).

Would it be possible to calculate absolute protein concentrations/copy numbers of the hits?

The use of stable isotope-labelled standards and rigorous assay validation are essential to establishing precise peptide concentrations. However, one important caveat is that the protein concentration is assumed to be equivalent to the concentration of the surrogate tryptic peptide measured. Accepting this assumption, one can calculate the protein concentration from the molecular weight of the surrogate peptide and the target protein. We added a brief section elaborating on protein concentrations to the Results and Discussion section (**p 17, lines 461-464**).

3. The selection of 2676 seems rather arbitrary and it is not clear which proteins were selected from the initial untargeted analysis?

We added more information on protein target selection to the Material and Methods section (**p 7, lines 196-205**). We further refer the reviewer to Supplementary Table 1, which lists the proteins identified in each tissue by the untargeted analysis (“Peptide ID by Untargeted MS”), followed by its selection and performance in initial assay development (“Response Curve Pass Y/N”). Briefly, if the assay did not pass the selection criteria, the target protein was excluded from the final panel. We clarify this in the manuscript (**p 14-15, lines 383-411**).

4. Which candidates were identified or selected by a relevant association to a disease such as the bridging factor 2.

All information on protein association with disease is available through the MouseQuaPro database. We also clarified the corresponding section of the manuscript (p 13, lines 354-365).

5. The authors selected 2676 proteins (page 10, line 290) for assay development and synthesized 2965 peptides. What is the average number of peptides used for MRM assay per protein? Overall, the authors developed 2118 assays. What happened with the remaining ~500 proteins?

The average number of peptides per protein is 1. The number of synthesized peptides was higher, since some of the selected peptides were found to be unsuitable for assay development (for example due to interferences or ion suppression from the sample matrix, or poor ionization properties). We refer the reviewer to Supplementary Table 1 for details on individual peptides which did not pass the validation process (columns “Response Curve Pass Y/N” and “Variability Pass”). We also clarified the difference in protein numbers in the manuscript (p 8-10, lines 231-278 and p 14-15, lines 383-411).

6. A weakness of the study is the initial number of identified proteins per tissue. So far, several studies have described protein expression in mouse tissues, and here the numbers based on untargeted MS are higher compared to the present study (Figure 1). For example, skeletal muscle tissue has been measured with ~500 proteins and in tissues such as spleen and skin ~1000 proteins have been identified. Even with in-solution digestion and the high-resolution fusion device used, more protein hits could be expected. The authors cited the Geiger publication. However, this work was from “2013” and used in-vivo SILAC, which results usually in less proteins hits due to the LysC digest compared to trypsin digestion. With a modern state-of-the-art MS one should expect definitely more hits.

The purpose of the untargeted analysis in this study was to identify targets for MRM-MS assay development, rather than exhaustively identify tissue proteomes. Therefore, our sample preparation followed a basic workflow that did not include techniques such as fractionation or depletion, which are used to increase the depth of proteome coverage at the cost of time and reproducibility. We avoided such techniques to ensure our sample preparation for untargeted MS analysis matched our sample preparation for MRM-MS. Additionally, for the protein identification step during analysis of the untargeted MS data, we required two peptides per protein. This increases our confidence in the identifications but lowers the overall total number of

identifications; a compromise which is essential for the intended application. We added this information to the Results and Discussion section (p 12, lines 343-346).

7. Finally, the Figure 3B summarized the number of assays and the skeletal muscle as well as the skin and testis reflect ~220 assays. It remains questionable whether this limited size of targets reflects a useful resource.

Skeletal muscle, skin and testis are difficult tissues to process and analyze, and the lower numbers represent this challenge. If additional specific targets are required, researchers could use our manuscript as a guideline for development of an MRM assay for their specific purpose (p 22, lines 529-531).

8. Moreover, the authors showed only Uniprot identifier in the SI data? Why is there no protein/gene name? That would be very helpful for users to find their target of interest.

We thank this reviewer for pointing out this oversight. For convenience of use, protein names were added to the supplementary tables, and both protein and gene names can be used to query the MouseQuaPro database.

9. In addition, the intensity and number of peptides in the untargeted analysis might be helpful data in combination with the MRM transitions.

We refer the reviewer to Supplementary Table 1, which lists the peptides identified in the untargeted analysis. The intensity values can be found in the data files uploaded to ProteomeXchange (PXD021333; see Data Availability statement p 11, lines 306-317).

10. SI table 1 is SI table 3. Table 2 shows the peptides hits of the untargeted analysis. It seems there is mixing of SI tables. SI table 3 is SI table 2.

We thank the reviewer for noticing this and have corrected the naming.

Moreover, the tables are far from a “detailed” summary. Why is it not possible to mention the protein names in the table? This is very inconvenient for the non-expert. The intensities of peptides, scores etc. are missing to better judge the quality of the data. The number of peptides per protein is also not listed. The summary of the protein group might be also helpful. Since muscle and heart show a high number of isoforms it would be nice to see whether all selected peptides are unique.

Again, we thank you the reviewer for this suggestion. We have added the Proteome Discoverer reports to the data uploaded in the ProteomeXchange repository so that complete search results

can be viewed. Protein names have been added to the supplementary tables. All additional information can be found in the MouseQuaPro database.

Reviewer #3 (Remarks to the Author):

#1 The study by Michaud et al provides a large scale of assays to MRM in 20 organs in multiple mouse strains. The quantification of each peptide has also been confirmed based on the CPTAC guidelines, which is a reliable result. Broad range of applications such as examination of biomarkers and proof of drug mechanism of action are expected.

#2 The authors have added assays for undetected proteins based on biological importance. (Line 285). The list of added proteins should also be shown separately from the set of detected proteins.

As described in the responses to reviewers #1 and #2 (points 3), we clarified the target protein selection in the Materials and Method section (**p 7, lines 196-205**). Additional biologically relevant information on the protein targets can be found in the MouseQuaPro database (**p 2, lines 46-48, p 11, lines 313-317, p 13, lines 363-365 and p 17, lines 446-450**).

#3 Figure 3 shows the number of peptide assays evaluated in each organ, but the numbers are quite different from the number of proteins identified for each organ shown in Figure 1. For example, plasma and erythrocytes are significantly increased compared to proteins identified from organs, but white fat, brown fat, epidymis, ovary, and testis appear to be decreased.

Is the increase in proteins due to the added biological significant proteins mentioned on line 285? For organs where the number of assays is reduced relative to the number of proteins identified, it would be preferable to state the reason for this, such as whether they are judged to be of low biological importance or because there are many peptides that are not suitable for quantification.

The number of plasma assays developed is larger than the number of detectable proteins due to additional proteins selected for biological relevance. This has been clarified in the text (**p 13, lines 354-365**). Furthermore, we clarified the protein selection in Material and Methods (**p 7, lines 196-205**).

#4 The authors develop assays in panels so that a total of 31 panels were designed with 1-4 panels for each organ sample. Each panel contains about 125 proteins (Line 355-356). The

authors should display the list of proteins covered by each panel and indicate which panel was used for which organ.

To clarify the panels used for each organ and tissue as well as which proteins are included per panel, a column has been added for each sample type in Supplementary Table 1. Please note that the separation into panels allowed us to efficiently measure all proteins across all 20 organs and tissues. It is by no means set in stone, and researchers can customize their panels within the sample of interest as needed (*e.g.*, liver peptides from panel 1 can be combined with liver peptides from panel 2). We only recommend that the described acquisition method parameters are maintained. We clarified this in the manuscript (**p 17, lines 435-441** and **p 22, lines 529-531**).

#5 Line 409, ‘The concentration of some SERPINA1a proteins further’, but this sentence is difficult to understand. Should it be ‘Serpina1 proteins’ rather than ‘SERPINA1a proteins’?

As suggested by the reviewer, the section on Serpina1 proteins was revised to correct the nomenclature (**p 19, lines 487-497**) and in Figure 5 (**p 20, below**).

#6 The authors should give the full names for RAG1 and RAG2 (Line 396).

We have updated the sentence to use the full names for RAG1 and RAG2 (**p 19, line 481-482**).

Figure 2. Concentration of serpin proteins in mouse organs and tissues. Mice have multiple *serpin1* genes, which vary according to mouse strain. We used five assays to measure cumulative and individual concentrations of Serpina1a-e proteins which demonstrate strain-specific expression patterns. Normalized concentration values are shown.

References

1. Mohammed, Y., Bhowmick, P., Michaud, S. A., Sickmann, A. & Borchers, C. H. Mouse Quantitative Proteomics Knowledgebase: reference protein concentration ranges in 20 mouse tissues using 5000 quantitative proteomics assays. *Bioinformatics* **btab018** (2021) doi:10.1093/bioinformatics/btab018.
2. Kustatscher, G. *et al.* Understudied proteins: opportunities and challenges for functional proteomics. *Nat Methods* **19**, 774–779 (2022).
3. Williams, T. I., Kowalchuk, C., Collins, L. B. & Reading, B. J. Discovery Proteomics and Absolute Protein Quantification Can Be Performed Simultaneously on an Orbitrap-Based Mass Spectrometer. *ACS Omega* **8**, 12573–12583 (2023).

REVIEWERS' COMMENTS:

Reviewer #3 (Remarks to the Author):

The authors have addressed my feedback, and the manuscript is now easier to understand. The validated MRM-MS assays that they developed are expected to monitor the disease progression of various disease model mouse and estimate the drug effect in vivo. I only have some minor comments.

#1 In the introduction section, it might be better to refer to "A multiplex protein panel assay for severity prediction and outcome prognosis in patients with COVID-19": An observational multi-cohort study" eClinicalMedicine 2022;49: 101495 to claim the usefulness of MRM assay.

#2 You should cite appropriate reports (p.3, lines 75-77, p.3, lines p.12, lines 343-345).

#3 It seems that N numbers of boxplots are not indicated in Figure 4 and Figure 5.

#4 You need to perform statistical analysis, especially for Figure 4. You said, "The four following immune-related proteins were significantly downregulated in the spleen of NOD/SCID mice" (p.19, lines 474)."

Reviewer #4 (Remarks to the Author):

This manuscript presents a valuable resource to the community and provides optimised and well-characterised targeted MS assays for >2000 proteins in 20 mouse tissues/organs.

Reviewer 2 raised the following issues.

1) the justification and comparison of the presented targeted assays with non-targeted assays. The authors have provided a reasonable rebuttal to this point, which I am satisfied with. The well-established and key attribute of targeted MS assays is the superior quantitative performance compared to untargeted MS approaches. A direct comparison of these approaches is not necessary as this has been established previously and is not the focus of the study.

2) More documentation about the MRM assays.

The authors have satisfactorily addressed this.

3) A weakness of the study is the initial number of identified proteins per tissue.

I agree with the author's rebuttal that an exhaustive proteomics characterisation of each tissue was not the purpose of the study but rather to develop a panel of assays to cover a significant portion of the proteome.

The authors have addressed the other minor points raised by reviewer 2, and I recommend that the manuscript is now suitable for publication.